# Using Deep Learning and Machine Learning Methods to Diagnose Hailstorms in Large-Scale Thermodynamic Environments

**Farha Pulukool, Longzhuang Li ***  **and Chuntao Liu**

Department of Computing Sciences, Texas A&M University-Corpus Christi, Corpus Christi, TX 78412, USA; fpulukool@islander.tamucc.edu (F.P.); Chuntao.Liu@tamucc.edu (C.L.)

**\*** Correspondence: Longzhuang.Li@tamucc.edu

**Abstract:** Hailstorms have caused damages in billions of dollars to industrial, electronic, and mechanical properties such as automobiles, buildings, roads, and aircrafts, as well as life threats to crop and cattle populations, due to their hazardous nature. Hence, the relevance of predicting hailstorms in the future has significant scientific, economic, and societal benefits. However, climate models do not have adequate resolutions to explicitly resolve these subscale phenomena. One solution is to estimate the probability of these storms by using large-scale atmospheric thermodynamic environment variables from climate model outputs, but the existing methods only carried out experiments on small datasets limited to a region, country, or location and a large number of input features. Using one year of Tropical Rainfall Measuring Mission (TRMM) observations and European Center for Medium-Range Weather Forecasts (ECMWF) Re-Analysis Interim (ERA-Interim) reanalysis on a global scale, this paper develops two deep-learning-based models (an autoencoder and convolutional neural network (CNN)) as well as a machine learning approach (random forest) for hailstorm prediction by using only four attributes—convective potential energy, convective inhibition, 1–3 km wind shear, and warm cloud depth. In the experiments, the random forest approach produces the best hailstorm prediction performance compared to the other two methods.

**Keywords:** hailstorms; hail models; hailstorms; hail predictions; deep learning

---

## 1. Introduction

In the context of climatic conditions, the term precipitation refers to a natural phenomenon where objects such as rain, hail, or snow precipitate from clouds. Water droplets collectively freeze to produce larger masses of ice called hailstones (or simply hail) in high-altitude regions with temperatures of less than 0 °C. The hail size grows when more super-cooled water drops merge and freeze on the hail core in strong updrafts in these clouds [1]. In due course, when the strength of the updrafts weakens, the heavier hailstones can no longer be sustained by the updrafts and are forced to plummet towards the Earth by gravity. From the time a hailstone is formed in a storm, its size grows or shrinks based on various physical processes that it encounters during its route towards the Earth from the storm [2]. The diameter of a hailstone can range from 0.2 to 6 inches. In the United States, hailstone sizes greater than 1–1.75 inches are considered harmful. Hailstones falling on the Earth move with different velocities based on their size. A hailstone with a diameter of 1 cm drops down to Earth at 20 mph, whereas an 8 cm hail plunges at 110 mph towards the Earth.

A hailstone is a solid precipitation that dives down to the Earth at a very high speed and can penetrate glasses of windows and houses, can easily shred vegetations like crops and trees, can thrash livestock and birds, and can crack roofs and walls. Winds can push hailstones sideways and cause

severe damage to anything lying in their way. Thus, hail is considered mostly catastrophic to multiple sectors, including agriculture, infrastructure, and transportation. The destruction caused by hailstones on crops, buildings and houses, vehicles, and roads causes them to be categorized as hazardous [3]. Hence, having a robust hailstorm prediction model that can identify and locate the occurrence of a hailstorm in advance is the best solution for alerting the public and reducing the damage impact.

Weather prediction is a highly complex and poorly constrained task due to the chaotic nature of Earth's weather components and the complexities involved in analyzing and processing meteorological data [4]. These complexities further increase during extreme events like hailstorms, tornadoes, and hurricanes, which are unforeseen climatic phenomena. Extreme events like hailstorms, floods, and hurricanes do not occur frequently, which makes their study more challenging, with fewer cases recorded. Extreme weather conditions and failures in proper sensor functioning can result in low quality of the captured data, presence of noise, and missing data. The sample size used for experiments can also be a determining factor.

Global climate imbalance is a major issue discussed on social platforms and has led to different active research and technologies in an effort to solve it. Significant prediction tools and models have evolved over the years, yet an accurate method or tool for predicting intense events like hail and lightning remains a huge challenge [5,6]. However, the last two decades have brought significant advancement in this field, driven by technological innovations. Balloon-borne videosondes for detecting high-altitude hail and calculating probable hailstone paths using various radar data, such as wind, speed, and direction, are a few state-of-the-art advances in hail predictions. Discriminating large hailstones from small ones using computer-based forecasting models that utilize data from dual-polarized radar can also be included in this category [7]. Numerical weather prediction models are the most commonly used forecasting tools; they gather current climatic conditions, process gathered data using computer models, and predict future weather conditions based on the collected data. In this approach, even slight discrepancies made by the model in capturing the initial weather condition to the fullest can cause erroneous forecasts [8].

Nowadays, it is well known that machine learning models can learn information directly from data without relying on a predetermined equation, and those models, especially deep-learning-based ones, have achieved great breakthroughs in many applications, including image classification [9], Android malware detection [10] and structural damage identification [11], to name a few. Gagne II et al. pointed out that machine learning forecasts have a higher critical success index (CSI) at most probability thresholds and greater reliability for predicting both severe and significant hail [2]. However, setting a threshold or criteria for meteorological data to determine the occurrence of an event using deep learning needs some thought. Meteorological data that keep changing every second with respect to space and time are difficult to process. For instance, several variables in the meteorological data determine an extreme event, like a flood or a hurricane. However, a machine learning application, like computer vision, that makes use of these steadily changing variables can execute at a rate much slower to the rate at which the actual event occurs. In addition, unlike existing methods that train their machine learning models using many attributes, this paper only utilizes four important attributes—convective potential energy, convective inhibition, 1–3 km wind shear, and warm cloud depth. One goal of this research is to investigate if a deep learning methodology still produces good results if given few attributes and a large amount of training data.

A dataset from the Tropical Rainfall Measuring Mission (TRMM) was used in this study. NASA and the Japan Aerospace Exploration Agency (JAXA) collaborated in launching the TRMM satellite for collecting active and passive microwave radiances from clouds and made huge contributions to many later research works pertaining to rainfall and storms [12]. The TRMM was launched containing five instruments with three sensors—a precipitation radar, microwave imager, and visible and infrared sensor—that provide information on clouds and precipitation. This information is processed into precipitation features that include properties of precipitation systems in Liu et al. [13] for 16 years from 1998 to 2013, before the satellite's mission ceased in 2015 (the radar stopped collecting scientific

data in September 2014) [12]. Combining the European Center for Medium-Range Weather Forecasts (ECMWF) Re-Analysis Interim (ERA-Interim) model data from Dee et al. [14], the 16-year database of precipitation features contains a great number of samples of storms with their remote sensing properties and their large-scale thermodynamic environments, which are suited to deep learning studies. This paper intends to design two deep learning models and a random forest model for hailstorm prediction. The predictions are validated by previous hailstorm events inferred by the TRMM. The TRMM precipitation radar provides the coverage only between 36S and 36N. It is well known that the TRMM dataset has been recognized as one of the best to describe global thunderstorms [15–18]. Earlier methods used for hailstorm predictions from Wang et al. [1], Gagne et al. [2], Kamangir et al. [8] and Marzbana and Witt [19], had a small dataset limited to a region, country, or location. The proposed method was used to carry out the experiment on a global scale and, hence, is a much larger dataset.

The major contributions of this paper are as follows:

(1)　The hailstorm prediction experiments are performed on a much larger dataset on a global scale, which covers one year of data in 2006 from the TRMM cloud and precipitation database with 2,030,007 records.
(2)　Two deep learning models—an autoencoder (AE) and convolutional neural network (CNN)—and one machine learning model—random forest (RF)—are developed and compared to predict hailstorms.
(3)　Only four attributes—convective potential energy, convective inhibition, 1–3 km wind shear, and warm cloud depth—are used in the experiments. The existing methods need to use a lot of attributes to produce good results.

This paper is organized as follows: Section 2 reviews the related works. Section 3 describes the proposed AE, CNN, and RF models. Section 4 describes the datasets as well as the performance comparison between the proposed models. Finally, the conclusion and scope for future works are described in Section 5.

## 2. Related Works

Like a biological neural network in the human body, an artificial neural network is a collection of interconnected neurons that are trained on tasks and learn to perform them efficiently through testing and validation. An artificial neural network comprises multiple layers, which also contain hidden layers. Input parameters and a constant weight in each layer determine the output of each layer. A few of the steps involved in implementing a neural network are as follows:

- Train a portion of the data (training data set) based on different input parameters.
- Optimize the model's complexity and validation of data.
- Evaluate a batch of data by testing it after training on a different batch of data.

The neural network problems can be divided into classification or regression problems. Regression models help determine the relationships among different variables in data, whereas classification problems classify data into different categories. The measure of error in a regression model can be calculated using the mean squared error function, while in a classification model, the cross-entropy function is generally used to measure the model's error rate. Some of the earlier methods for hailstorm prediction are specified in the upcoming section.

### 2.1. Bayesian Neural Network Model for Hailstorm Prediction

Marzbana and Witt [19] proposed a Bayesian neural network (BNN) model, which is a neural network designed to function based on statistics and utilizes a probabilistic approach for predicting hail sizes. Both classification and regression models with a sample size of 386 were used for this study. The regression model plotted its predictions in a scatter plot, showing actual versus predicted values of hail sizes. On the other hand, the classification model used Heidke skill scores (HSS), a receiver

operating characteristic (ROC) diagram, and contingency tables to represent the predicted results [19]. The probability function utilized in the Bayesian model helped determine optimum input parameters that could produce accurate hail size predictions.

### 2.2. CNN Models Using Image Processing for Hailstorm Prediction

Over the last few years, image processing techniques have made significant contributions in weather prediction. Wang et al. [1] proposed a convolutional neural network model that classifies hailstorms using multiple images as input. A CNN model that uses images to predict hail for four different provinces in China was proposed as a recent advance in hail prediction using deep learning. Hailstorm cases between the years 2005 and 2016 were considered for the study. A good understanding of meteorological data and neural network models is quite essential in designing a CNN for hail prediction. According to the model proposed by Ref. 1, a high radar echo above −20 °C with an overhanging structure can be categorized as hail. Data preprocessing in this CNN model is a three-step process, as shown in Figure 1.

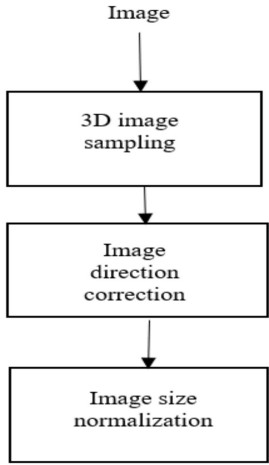

**Figure 1.** Data preprocessing.

The input of the CNN is a three-dimensional image grid of the data that are sampled. The next step can be called image preprocessing. Here, it corrects the direction of the echo in a hailstorm image using the optical flow method. Following this, the intensity values of different pixels in the image are altered so that they improve the contrast. This process is called image normalization. Finally, a CNN model with eight layers is designed to avoid overfitting and underfitting. The CNN model in Wang et al. [1] yielded more true positives and negatives and less false positives and negatives when compared to the results obtained from the probability of severe hail (POSH) method. However, the size of the dataset is relatively small, making it difficult to evaluate the accuracy of the model. Using a convolutional neural network optimized by parameters has turned out to be a promising prediction method due to the higher accuracy rates. The CNN specified in Maskey et al. [20] uses a training data set consisting of previous instances of hailstorms, along with their respective Next Generation Weather Radar (NEXRAD) images. The CNN classified the images where hail was present. It was also pointed out that most of the hailstorm prediction techniques require efficient preprocessing of the data, which can be complicated. Hence, Maskey et al. [20] utilized a CNN optimized by parameters that has high accuracy compared to the existing CNN models.

### 2.3. Features and Target Forecasting of Hail with Machine Learning

Conventional hail forecasting methodologies make use of models based on the principles of physics, which, in turn, depend on mathematical equations, hypotheses, and an event's start and end conditions. On the other hand, machine learning models process a group of data, analyze them, try to

find out the relationships among various attributes or features of the respective dataset, and then estimate the accuracy or error rate of their observed relationship pattern. Gagne II et al. [21] present a machine learning (ML) hail regression model that can forecast hail one day in advance. This regression model is an improvement of a previous hail prediction model proposed in Gagne et al. [22]. This new model aims to predict probable maximum hail size distribution in a certain region of a storm. Further, Gagne II et al. [21] compared the predictions of severe hails from a deep learning model with a machine learning model (ML). This ML model makes use of ensemble systems of two convection-allowing models (CAMs) to test the accuracy of its hail predictions. A set of weather predictions, as opposed to a single prediction, is termed as ensemble forecasting. Initially, the model does data processing, which involves identifying and locating storms; then, it extracts data and performs forecasts over selected regions.

The main process in Gagne II et al. [21] is using all important variables that define the storm to predict hail and forecast a distribution of hail. A classifier that identifies the occurrence and non-occurrence of hail and a regressor that predicts the distribution of hail occurrences were the key operations used in this study. A random forest classifier that distinguishes between a hail and non-hail, as well as an elastic net regressor or random forest regressor that predicts the hail distribution, carried out the main functionality of the model. The sampling was done over the United States in this model. The accuracy of the model was verified using different criteria, such as ROC curves, reliability, performance and attribute diagrams, and their respective metrics. Overall, this machine learning model showed a high success rate in classifying hail from non-hail; however, the prediction of more false positives in hail occurrence is a downfall of this model. Another drawback of this model is that the maximum hail sizes predicted were undercut, and there were errors in spatial and temporal domains with respect to the storm location.

## 3. Proposed Method

In this section, two deep learning models, AE and CNN, as well as a machine learning model (RF), are presented.

### 3.1. Features and Target

The dataset used for this study is the TRMM cloud and precipitation database. The database has been classified into three levels based on the data features, source, and purpose. We make use of level 2 data in this paper. Level 2 data define precipitation features and are saved in an HDF format. Liu [23] explains the classification and description of the TRMM precipitation and cloud feature database in detail.

A feature in a database refers to a measurable entity that can be used to analyze an object. The features selected in this study are variables from level 2 of the TRMM dataset in Liu et al. [24]. Convective available potential energy (ERA_CAPE), convective inhibition (ERA_CIN), and wind shear (ERA_SHEAR13KM) are the important thermodynamic environment parameters considered in a convective storm. We choose the aforementioned attributes primarily in the training and testing sets; additionally, the ECMWF reanalysis total column water vapor (ERA_TCWV) is used as an input parameter to monitor the changes in predictions for this study. Other features that are not included in the training and testing sets are latitude, longitude, year, temperature in degrees Celsius at maximum 40 dBZ height, and number of pixels at 40 dBZ height. Liu [23] explains the above features in detail.

The target of this paper is to determine the presence of hailstorms in the TRMM precipitation features between 1998 and 2013 using deep neural network models only based on the thermodynamic environment variables from ERA-Interim reanalysis. The results of the predictor are validated by comparing them to the hailstorm inferred by TRMM high radar echoes reaching high altitudes

(temperature < −22 °C) during the respective years based on the data gathered by the TRMM satellite. The target can be described in Equation (1):

$$H_y = \begin{cases} 0, & R < 40 \text{ dBZ at } T < -22\,°\text{C} \\ 1, & R \geq 40 \text{ dBZ at } T < -22\,°\text{C} \end{cases} \tag{1}$$

where *R* is the maximum radar reflectivity from the TRMM precipitation radar (PR), and *T* is the temperature.

Ni et al. [3] state that, based on the studies conducted using passive microwave radiometers and precipitation radar, any precipitation features with reflectivity of 44 dBZ at temperature −22 °C or above have a high probability related to the ground hail reports over the US. The differences between 40 dBZ used in this paper and 44 dBZ, as used in Ni et al. [3], are understood. This serves as a criterion that this paper sets forth in distinguishing hail from 'non-hail'. Additionally, a random forest model is used in this study for predicting hailstorms based on the same criteria, and the accuracies of the deep neural network models and random forest predictions are compared and analyzed.

*3.2. Methodology*

Two deep learning architectures are developed in this paper: an AE model and a CNN model. The AE model gives a better representation of the original input through lossy compression. In addition, the AE model carries out denoising and reconstructs the raw input. This input is passed into a binary classifier, which predicts whether the output is a hailstorm (1) or non-hailstorm (0). This study implemented an AE, which consists of an encoder layer and a decoder layer of 32 neurons each (see Figure 2). The binary classifier of the AE model consists of two fully connected layers and uses 'softmax' as the activation function. The second model is a CNN (see Figure 3), which is a widely regarded approach for classification problems. When it comes to capturing spatial and temporal dependencies, CNNs perform very well, with minimal data preprocessing required. Although CNNs are mostly implemented in image processing, they can be used for other data types as well. The CNN model developed in this paper comprises two layers with 64 and 32 filters, respectively. Two fully connected layers and 'softmax' were adopted in the output layer of the CNN model. The third model is an RF classifier that is based on a set of decision trees. All these models were tested with three input parameters, namely ERA_CAPE, ERA_CIN, and ERA_SHEAR13KM. ERA_TCWV was also used as a parameter for testing purposes.

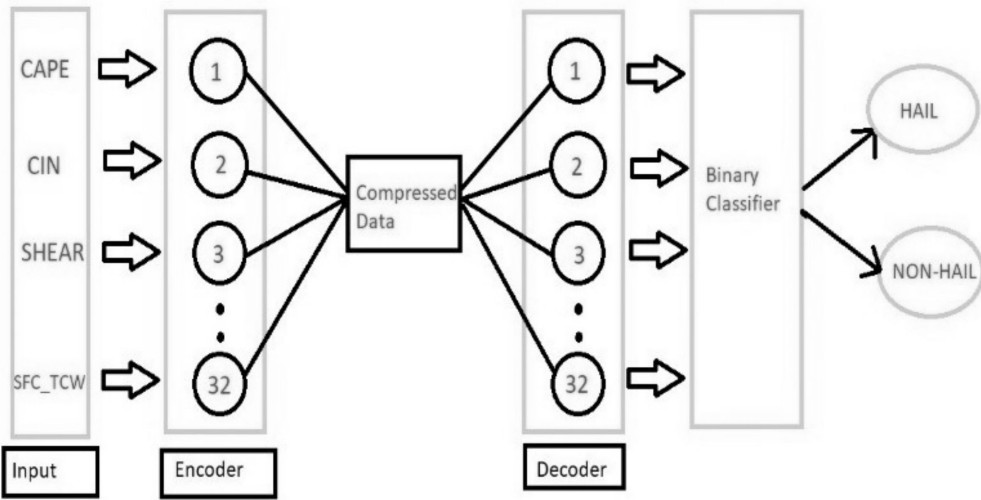

**Figure 2.** The proposed autoencoder model.

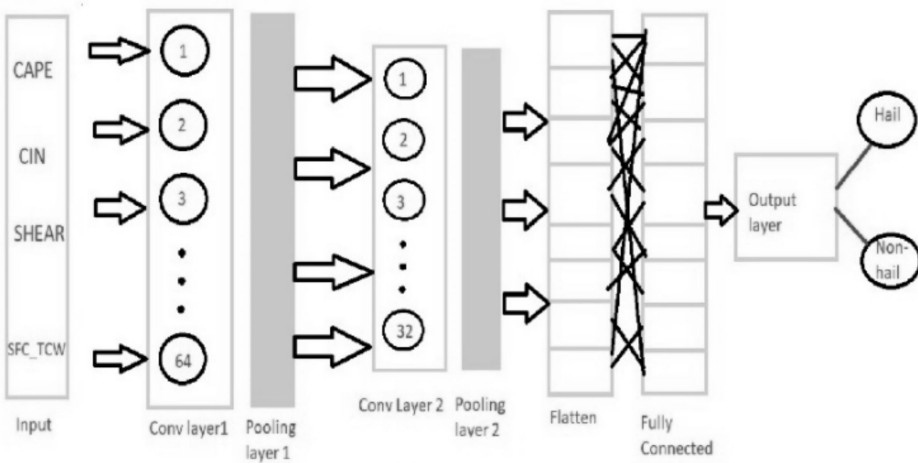

**Figure 3.** The proposed convolutional neural network (CNN) model.

The activation functions used in the deep learning models are specified in each respective layer added. There are several activation functions available, such as selu, tanh, softmax, linear, softplus, softsign, sigmoid, relu, hard-sigmoid, exponential, and others. We mostly make use of 'tanh' and 'relu' as well as Max pooling in the convolution layers and 'softmax' in the output layer. Once the input goes from the initial layer and reaches the output layer, the respective classifier of the AE or CNN makes a target of two classes, 0 and 1. From the predictions made by the model, the highest value of the two classes is chosen as the final output of the binary classification for each input value. In the next step after a model definition, the dataset is fed and split into training and testing sets. For this experiment, the dataset was split into 70% training and 30% testing. The latitude and longitude values of the true test sets ($y\_test = 1$) are copied to a dictionary for plotting the map later. Data preprocessing is essential for standardizing the input and obtaining efficient outputs. For this purpose, standard scalar is used in normalizing the data. The next step was to convert the labeled variables in the data into a format that can be understood by the deep learning model and enhance its accuracy in predictions. Label encoding and one hot encoding enable this conversion of data variables into a model-interpretable format.

Then, the model was compiled, and the training phase followed. By training the model, the inputs are matched to the outputs by finding the best sets of weights. Network weights in the training dataset are updated using the Adam optimization algorithm. Training was performed on selected batches of data mentioned by batch size. After each batch was trained, the model was expected to improve its predictions, minimizing errors. MSE (mean square error) was the loss function used. While training, the number of epochs should be specified, where epoch indicates the number of iterations done through each row of the entire training dataset. Making predictions on the test dataset using the respective model was done after training. Predictions for the remaining test dataset yield an output of either 1, indicating the presence of a hailstorm, or 0 otherwise. The 'softmax' function was chosen in the output layer due to its high performance in classification problems.

## 4. Model Performance Evaluation

The models were coded using python 2.7, Tensorflow, and Scikit-learn. The AE and CNN models were run for between 5 and 20 epochs. with each batch size set to 32. The RF model used a combination of only 10 different decision trees for its predictions. Unlike earlier models comprising a smaller dataset and large number of input features selected, our models were trained and tested on a global scale using one year of data in 2006 from the TRMM cloud and precipitation database with 2,030,007 records, out of which 1,421,004 PFs were used in training and 609,003 for testing. In addition, only four input parameters were chosen to determine hailstorms. In the AE model, the learning rate was set as 0.01, the activation function of the encoder was *linear*, the activation function of the decoder was *tanh* (hyperbolic tangent), and the hidden layer was 1. In the CNN model, the learning rate was set as 0.01,

the activation functions of the first and second convolution layers were *linear* and *tanh*, respectively, the kernel size was $3 \times 3$, and the activation function of the dense layer was *softmax*.

A classification report and confusion matrix were generated in order to evaluate the performance of each model. The confusion matrix categorizes the predictions as true positives (TPs), true negatives (TNs), false positives (FPs), and false negatives (FNs). A classification report has different scores indicating how good the predictions are, and the confusion matrix displays the number of hits, misses, false alarm rates, and correct rejection rates. The precision, recall and F1 score for hailstorm or no-hailstorm prediction were computed as follows:

Precision: It is the ratio of the correctly predicted hailstorm (or no-hailstorm) result to the total hailstorm (or no-hailstorm) prediction.

$$Precision = \frac{TP}{TP + FP} \tag{2}$$

Recall: It is the ratio of correctly predicted hailstorms (or no-hailstorms) to total actual hailstorms (or no-hailstorms).

$$Recall = \frac{TP}{TP + FN} \tag{3}$$

F1 score: A measure of a test's accuracy by considering both the precision and the recall.

$$F1 = 2 * \left[ \frac{precision * recall}{precision + recall} \right] \tag{4}$$

The probability of detection (POD), false alarm rate (FAR), and critical success index (CSI) of the models were also computed. POD is the same as recall in Equation (2). The formulas for computing FAR and CSI are shown below:

$$FAR = \frac{FP}{TP + FP} \tag{5}$$

$$CSI = \frac{TP}{TP + FN + FP} \tag{6}$$

### 4.1. Experimental Results

Next, the experimental results with the AE, CNN, and RF models are shown. Although the models were executed 10 times, only the best results are presented. The results are displayed on two panels; the top panel shows the test values expected to be obtained, and the bottom panel shows the predictions made by the respective models. A map showing the autoencoder model's results is shown in Figure 4, where the x-axis is the longitude and the y-axis is the latitude (x-axis and y-axis are similarly defined in Figures 5 and 6).

The predictions made by the AE model are considerably less than the expected values shown by the test set in the top panel. The model was not able to identify prime spots of hailstorms, except for a few. Table 1 presents the confusion matrix of the AE's predictions, and Table 2 depicts the performance scores of the respective predictions made. Table 1 signifies that there are very few true positives, and the number of predicted false positives is big when compared with predicted true positives. However, the model was able to classify most of the no-hailstorms as true negatives, but the model's capability to identify hailstorm classes was weak. It is no surprise to see in Table 2 that the precision, recall, and F1 score for no-hailstorms are close to one and are very low for hailstorm prediction.

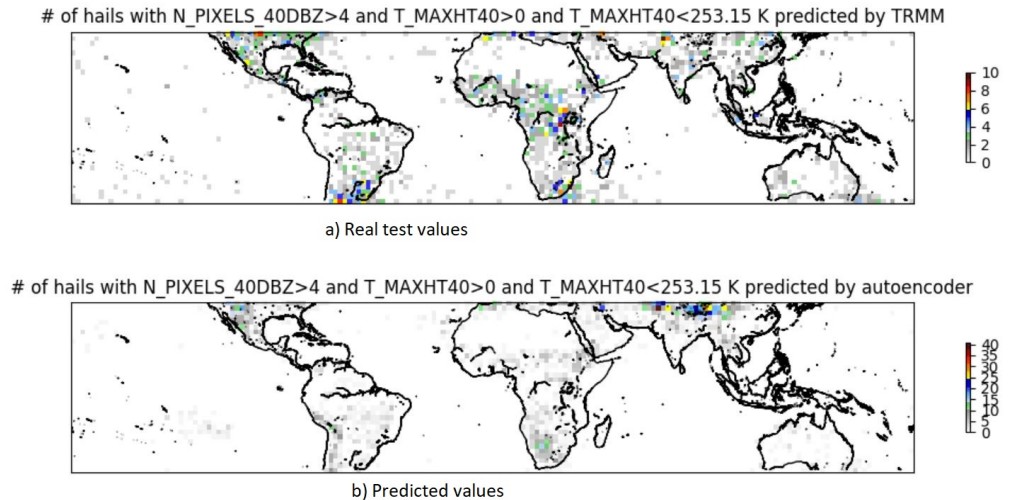

**Figure 4.** The hailstorm test values and predictions for 2006 using the autoencoder (AE) model. (**a**) Real test values. (**b**) Predicted values.

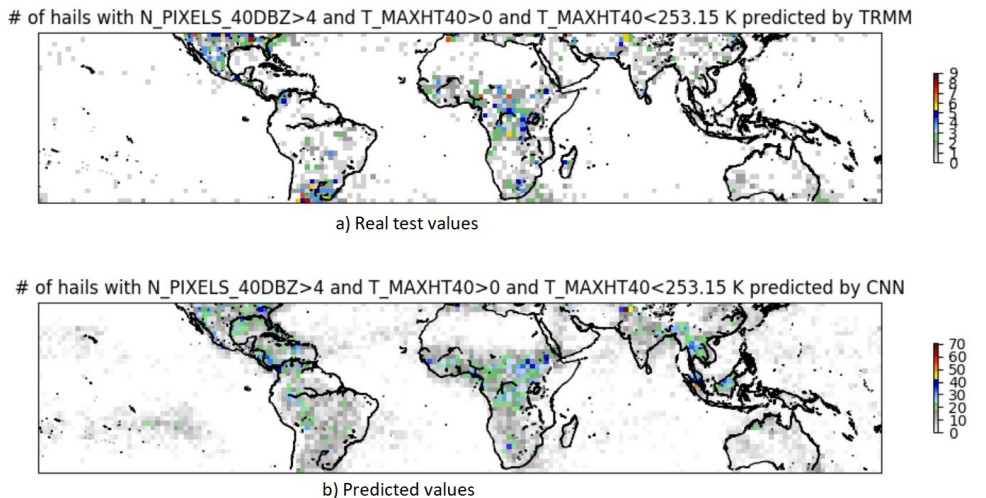

**Figure 5.** The hailstorm test values and predictions for 2006 using the CNN model. (**a**) Real test values. (**b**) Predicted values.

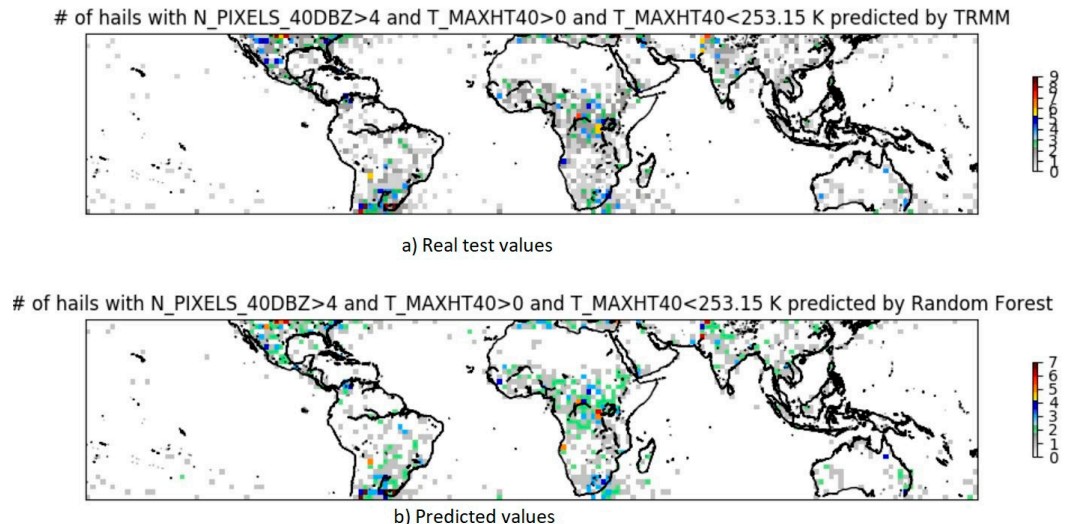

**Figure 6.** The hailstorm test values and predictions for 2006 using the random forest (RF) model. (**a**) Real test values. (**b**) Predicted values.

**Table 1.** Confusion matrix of the predictions based on the AE model.

|  | Predicted No-Hail | Predicted Yes-Hail |
|---|---|---|
| Real no-hail | 604,316 | 2896 |
| Real yes-hail | 1675 | 116 |

**Table 2.** Performance measurements of the AE model.

|  | Precision | Recall | F1 Score |
|---|---|---|---|
| No-hail | 0.997 | 0.995 | 0.996 |
| Hail | 0.039 | 0.065 | 0.049 |

The predictions of the CNN model as well its respective confusion matrix and performance scores are shown in Figure 5 and Tables 3 and 4, respectively. According to Figure 5, the predictions are also scattered over the ocean, which is misleading from the actual test values. In Table 3, the CNN model accurately classified a greater number of hailstorms; however, the number of false positives is significantly large when compared to the AE. Table 4 suggests that the CNN model scored slightly better when compared to the AE model. For instance, the 'pod' of AE was 6% and the CNN improved its 'pod' to 40%, which entitles the CNN to have a higher precision. Nevertheless, having a high 'far' reduces the efficiency of the model.

**Table 3.** Confusion matrix of the predictions based on the CNN model.

|  | Predicted No-Hail | Predicted Yes-Hail |
|---|---|---|
| Real no-hail | 584,159 | 23,053 |
| Real yes-hail | 1058 | 733 |

**Table 4.** Performance measurements of the CNN model.

|  | Precision | Recall | F1 Score |
|---|---|---|---|
| No-hail | 0.998 | 0.962 | 0.980 |
| Hail | 0.031 | 0.409 | 0.058 |

The plot, confusion matrix, and performance measures of the RF model are illustrated in Figure 6 and Tables 5 and 6, respectively. The RF predictions were almost identical to the actual test values. The hailstorm distributions worldwide were captured with high accuracy using the RF (see Figure 6). Only a slight variation is visible in the number of hailstorms classified by the RF model. The number of false positives in Table 5 is very low, and just a few hundreds of false negatives were obtained in the confusion matrix. The storms were classified with close resemblance to the expected test values. Table 6 clearly suggests that the results from the RF model markedly outperformed the other two deep learning models, with high skill scores in all the performance criteria.

**Table 5.** Confusion matrix of the predictions based on the RF model.

|  | Predicted No-Hail | Predicted Yes-Hail |
|---|---|---|
| Real no-hail | 607,200 | 12 |
| Real yes-hail | 502 | 1289 |

**Table 6.** Performance measurements of the RF model.

|  | Precision | Recall | F1 Score |
|:---:|:---:|:---:|:---:|
| **No-hail** | 0.999 | 1.000 | 0.999 |
| **Hail** | 0.991 | 0.720 | 0.834 |

*4.2. Comparison with Existing Methods*

The skill scores of existing methods and the proposed models are summarized in Table 7. The skill scores of proposed AE and CNN models are relatively low with respect to the earlier models. Both models have a high FAR score, and the POD scores are far below expectations. However, the RF model has a good POD score, and the FAR is negligible with respect to the other models. When it comes to CSI, the RF model has again shown greater accuracy. The datasets used in the earlier models [1,8] are relatively very small compared to what the AE, CNN, and RF models used in this paper. For example, the CNN model from Ref. 1 only used the data for 381 storm days in one region. In addition, most of the earlier models used large numbers of input features; for example, the stacked denoising autoencoder (SDAE) in Kamangir et al. [8] used 32 features, which can be decisive in predicting. The proposed model used no more than four features in that regard. Hence, the proposed AE and CNN models' performances cannot be compared solely based on these results. Overall, the RF model yields promising results with its good POD value and second highest CSI value, as well as a very negligible FAR value.

**Table 7.** Skill scores of existing methods and proposed models.

| Model | POD % | FAR % | CSI % |
|:---:|:---:|:---:|:---:|
| CNN (Wang et al., 2018) | 80 | 10 | 74 |
| POSH (Wang et al., 2018) | 73 | 39 | 49 |
| SDAE (Kamangir, 2019) | 86 | 40 | 57 |
| CT2016 (Kamangir, 2019) | 92 | 93 | 7 |
| NDFD (Kamangir, 2019) | 86 | 93 | 7 |
| PCA (Kamangir, 2019) | 82 | 94 | 5 |
| Proposed AE | 7 | 96 | 2 |
| Proposed CNN | 41 | 97 | 3 |
| Proposed RF | 72 | 0.9 | 71 |

## 5. Conclusion and Future Works

This paper developed two deep learning models and a machine learning model that can be applied to large-scale thermodynamic environment variables in the diagnosis of hailstorms globally. The AE and CNN models can predict a good number of true negatives, but both models made predictions with low precision and, moreover, with high error rates. Nevertheless, the proposed random forest model is robust and reliable with a high accuracy score, and it made most of its predictions true positives. Unlike earlier models that used many input features over small datasets, the proposed deep learning models performed well with just four input features and a huge dataset. However, the random forest model had a superior performance compared to the proposed deep learning models.

There are many thermodynamic environment variables that may play an important role in creating hailstorms. Here, we have chosen a few of the most important parameters that have been used in the past literature [25,26]. However, it takes a significant amount of input data to train the model, and hailstorms are relatively rare (we have a relatively small target compared to the total sample size). We could add more variables in the future when more data become available from Global Precipitation Measurement (GPM) satellites. In addition, it could be worthwhile to test different sets

of input variables. This paper aims to demonstrate a methodology rather than to provide the best solution. Further investigation on this valuable approach is warranted for the future.

Potential future works include improving the performance of the AE and CNN deep learning models by using more convolution layers. The proposed deep learning model predictions can be validated with the CAM ensemble system in Gagne et al. [21] to determine the accuracy, and the model architecture can be changed accordingly. Other deep learning models based on gated recurrent unit (GRU) or long-short term memory (LSTM) can be developed to see if the false alarm rates are lower compared to the AE and CNN. Similarly, more emphasis on data preprocessing can be given in future models.

**Author Contributions:** Conceptualization, F.P., L.L. and C.L.; methodology, F.P. and L.L.; software, F.P.; validation, F.P., L.L. and C.L.; formal analysis, C.L.; resources, C.L.; data curation, F.P. and C.L.; writing—original draft preparation, F.P.; writing—review and editing, L.L. and C.L.; supervision, L.L.; project administration, L.L. All authors have read and agreed to the published version of the manuscript.

**Funding:** This research received no external funding.

**Acknowledgments:** The authors would like to acknowledge Thomas Lavigne (TAMU-CC) and Lindsey Hayden (TAMU-CC) for the constructive discussions on data processing and manipulation.

**Conflicts of Interest:** The authors declare no conflict of interest.

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
