# Peer review of "Using Deep Learning and Machine Learning Methods to Diagnose Hailstorms in Large-Scale Thermodynamic Environments"

_sustainability, doi:10.3390/su122410499_

Round 1
Reviewer 1 Report
This study attempts to classify hail storms from a global reanalysis product utilizing two deep learning models and one machine learning model. The models are trained and compared to results from a satellite analysis product for the year of 2006. Only four inputs are utilized in these networks which, if model performance is good, is a useful characteristic of a model. However, in this study the models appear to perform relatively poorly compared to other studies involving machine or deep learning for hail prediction. This is not to say that this study is not needed or helpful, but many of the results and conclusions make it seem like the models show improvement. At best, the RF model is on par with other studies and there appears to be an error in the table summarizing the results (Table 7, FAR% - see comments below). I recommend several revisions throughout the paper below, but specifically within the introduction, results, and conclusions sections.
Paragraph starting on line 55 - This paragraph is confusing and does not read well:
- Modeling is brought up as the most commonly used, then deep learning is said to be the preferred method, then the paragraph finishes with several reasons why deep learning is problematic.
- This paper started with an extremely basic introduction into hail but within this paragraph assumes a working knowledge of deep learning.
- The use of "geographical data" and "geoscience data" is vague - if this is referring to meteorological data (which does change rapidly in space and time) and not just any data with a reference location, why not use "meteorological data" instead?
- Overall, this paragraph may need a fair amount of changes.
Line 295 - I don't think this is explained correctly. TP (yes-yes) is very small, FP and FN are much larger than TP (but the text says FP is small as well... 116 vs 2896 is a large difference - are you referring to it as small relative to the full dataset?). Then on line 297 it says the model was able to correctly classify the TPs which was already stated as being very poor. I assume this is supposed to say true negative, but in general this discussion requires cleanup.
Table 2 - are the no-hail values correct? "Precision" for the no-hail should be TN/(TN+FN), or the correctly predicted no-hailstorms divided by the total predicted no-hailstorms. It seems here you are taking TN/(TN+FP): 604316/(604316+2896) = 0.995 where it should be 604316/(604316+1675) = 0.997. I have not checked the stats in all of the tables but highly encourage the authors to verify the tables are being populated correctly as another possible mistake in Table 7 is found and discussed below.
Figures 4, 5, and 6 - the paper mentions that this is over a global domain but the images here are only showing between maybe +- 35Ëš latitude which excludes areas of the very high concentrations of hail (specifically the US and Eurasia). I'm assuming this is just because this is the coverage of the TRMM dataset, and if this is the case, is TRMM the best observational dataset for this type of problem? Please explain in the text. Additionally, please add axes labels.
Results section -
- Table 7 shows POD percentages that don't match the precision from the tables above. How are these POD percentages calculated? The paper says they are from "many experiments" but what does that mean?
- The RF model is said to have a high POD percentage but, excluding the other two models from this study, it is the poorest performer of the other models shown.
- The FAR% for the RF model is extremely small. When you take the values from Table 5, it appears that maybe the value was not multiplied by 100 as the others are to get a percentage (i.e., 12/(12+1289) = 0.0092 which, to get to a percentage, needs to be multiplied by 100 to give 0.9%).
- The use of accuracy in this type of situation seems misleading - the number of non-hailstorms is much larger than the number of hailstorms, thus, including TN in a model in which TN is orders of magnitude larger than TP (which is what actually causes damage and needs to be predicted) makes these models seem like they are performing well. CSI is the more responsible metric.
Line 358 - the AE and CNN were poor performers but the conclusions say they predict a "good" number of true positives... the recall scores show that the number of true positives relative to the number of observed hail pixels is poor for both models. How can the conclusions state that these are good?
Lines 359 and 362 - see above comment about the use of "accuracy" as a metric in these scenarios
Line 362 - most of the predictions were "true negative" and not "true positive"
Line 364 - it is generous to say these models performed well compared to the others. It seems that they may be missing key meteorological inputs to get this right on top of the other suggestions for improvement. Why is it that these four variables are the only ones tested? One of the benefits to using a numerical weather prediction model for training deep/machine learning models is that there is full spatial coverage of a lot of variables? Can the use of additional variables improve these results further? A physical understanding of hail formation would lead one to assume that there is more than CAPE, CIN, shear, and integrated water vapor to produce hail.
Misc. comments:
Line 52 - "conv4 has different accuracy": not sure this belongs here.
Line 69 - ")" should be deleted.
Line 113 - List should have bullets and capitalization
Line 167 - define "CAM"
Line 199 - Should not use variable names as they appear in the dataset... give the real names of these variables: "LAT, LON, YEAR, T_MAX_HT_40 and "N_PIXELS_40DBZ"
Reviewer 2 Report
This manuscript employed deep learning and machine learning techniques to develop the climate models to predict the hailstorms from large-scale thermodynamic environments. The experimental data were used to validate the performance of the proposed prediction models. The results shown that the random forest method outperform AE and CNN in terms of prediction accuracy. Overall, the topic of this research is interesting, and the manuscript was well organized and written. My detailed comments are provided as follows.
- Please illustrate the main innovation of this study. Why were machine learning methods considered for task of interest?
- When the CNN method was introduced, please consider the following reference to be included.
https://doi.org/10.1177/1475921718804132
- Please give more details on how to assign the hyperparameters of the learning models.
- More future research should be included in conclusion part.
- There are several obvious typos in the abstract: “prop-erties” should be “properties”, “reso-lutions” should be “resolutions”, “da-taset” should be “dataset”, etc.
